

# Detecting cloud contamination in passive microwave satellite measurements over land

Samuel Favrichon[1], Catherine Prigent[1], Carlos Jimenez[2], and Filipe Aires[1]

[1]Sorbonne Université, Observatoire de Paris, Université PSL, CNRS, LERMA, Paris, France
[2]Estellus, Paris, France

**Correspondence:** Favrichon Samuel (samuel.favrichon@obspm.fr)

**Abstract.** Multiple geophysical parameters such as land surface temperature, are estimated using Microwave (MW) remote sensed brightness temperature. It is known that clouds do not affect those measurement in the MWs as much as in Visible and Infrared (VIS/IR), but some contamination can still occur when strong cloud formation (i.e. convective towers) or precipitation are present. To limit errors associated to cloud contamination in the estimation of surface parameters, we build an index giving

the confidence to have an observation clear from contamination using standalone MW brightness temperature measurements. The method developed uses a statistical neural networks model built upon the Global Precipitation Microwave Imager (GPM-GMI) observations, with cloud presence information taken from Meteosat Third Generation-Spinning Enhanced Visible and Infrared Imager (MSG-SEVIRI). This index is available over land and ocean, and is developed for multiple frequency ranges to be applicable to successive generations of MW imagers (10 to 40 GHz, 10 to 100 GHz, 10 to 200 GHz). The index confidence

increases with the number of channels available, and performs better over the ocean as expected. In all cases, even with a reduced number of information over land, the model reaches an accuracy $\geq 70\%$, in detecting contaminated observations. Finally an example application of this index to eliminate grid cells unsuitable for land surface temperature estimation is shown.

*Copyright statement.* TEXT

## 1 Introduction

Visible/Infrared (VIS/IR) satellite imagers provide excellent information about land surface characterization. The applications include land surface temperature estimations (e.g. Freitas et al., 2013), vegetation characteristics (e.g., Tucker et al., 2005), or surface water extent (e.g., Pekel et al., 2016). These geophysical parameters can be retrieved accurately and with a good spatial and temporal resolution from VIS/IR observations, but only under clear sky conditions. With clouds covering $\sim 60\%$ of the globe at any time (Rossow and Schiffer, 1999), there is a need for alternative sources of information. Passive microwave

observations from satellites can partly fill this gap: they are much less sensitive to clouds and can provide valuable estimates of the surface properties, despite their coarser spatial and temporal resolution. Today, land surface temperature can be retrieved from IR observations for $\sim 60\%$ of the locations with a spatial resolution of 1 km twice a day from polar orbiters (Prata et al.,



1995) and with a spatial resolution of $2\,km$ every 15 minutes from geostationary satellites (e.g., Schmit et al., 2017). On the other hand, passive microwaves can provide this information with a spatial resolution of $\sim20\,km$ spatial resolution twice a day over $\sim100\%$ of the continents (Aires et al., 2001). Programs are underway to merge these different observations for a complete spatial and temporal coverage. For instance, long time series of land surface temperature estimations with passive

microwave observations are under-construction, using different generations of passive microwave satellite instruments to be used in synergy with IR estimates (e.g., Prigent et al., 2016; Jiménez et al., 2017).

Although microwaves are less sensitive to clouds, the effect of clouds and rain on the microwave radiation increases with frequencies. Multiple effects can occur, from liquid water clouds and rain emitting passive microwave radiation at the physical temperature of the cloud or rain, to scattering by ice clouds that can lower the measured brightness temperatures especially at

high frequencies and for large ice contents. The cloud / rain effect that can be detected strongly depends upon the surface type. The surface contribution to the passive microwave observations is proportional to the surface emissivity that changes from $\sim0.5$ over ocean to $\sim1$ over dry soil or dense forests. This means that the contrast between the liquid particles in the cloud and rain and the surface will be usually larger over ocean than over land: the cloud and rain liquid water emission increases the brightness temperature over the radiometrically cold ocean but will not show much contrast over the already radiometrically

warm land. The opposite will prevail for frozen clouds, with the cloud scattering depressing the brightness temperature above a radiometrically warm land surface. Over ocean, passive microwaves have been extensively used to quantify the cloud liquid water and rain amounts (e.g., Greenwald et al., 1993; Kummerow et al., 1998). For ocean surface applications, the cloud liquid water amount can usually be accounted for and the surface parameter estimation can compensate for the cloud impact, when the atmospheric transmission is still high enough to have a significant contribution from the surface. Over land, cloud and rain

detection using passive microwave is much more complicated (e.g., Spencer et al., 1989; Aires et al., 2001). First, the surface emissivity is usually close to one reducing the contrast between the cloud and the surface and second, it changes spatially and temporally, with e.g., variations in soil moisture, vegetation density, or snow cover (e.g., Prigent et al., 2006). This can seriously affect the retrieval of land surface parameters when a cloud or rain effect is miss-interpreted as a surface change.

The objective of this study is to develop a method that indicates the cloud / rain contamination on the passive microwave

(MW) observations over land, for different ranges of frequencies available on board the successive generations of passive MW satellite instruments. Rain detection schemes have been developed for the Special Sensor Microwave / Imager (SSM/I) over land: they are based on the scattering signal at $85\,GHz$ and use decision trees (Grody, 1991; Ferraro, 1997). Cloud filtering methods have also been derived, for specific applications or for a given instrument. Long et al. (1999) analyzed the brightness temperature time series at $85\,GHz$ with different methods to remove the cloud perturbation on the SSM/I images

for land surface applications. For the estimation of upper tropospheric humidity with satellite measurements around the water vapor line at $183.31\,GHz$, Buehler et al. (2007) developed filters with different channels around the line to avoid the cloud contaminated grid cells. Aires et al. (2011) used a neural network method trained on Meteosat Third Generation Spinning Enhanced Visible and Infrared Imager (SEVIRI) cloud products to create a cloud mask and classification from the Advanced Microwave Sounding Units A and B (AMSU-A / AMSU-B) with channels from $23\,GHz$ to $183\,GHz$: statistical models were

built separately over land and ocean to detect clouds or classify them into clear sky, low, medium, or high clouds.



Here, we use a similar approach to Aires et al. (2011) to develop a cloud / rain indicator over land, for the passive MW imagers used for the estimation of land surface parameters over the last decades. Starting from early 80's with the Scanning Multichannel Microwave Radiometer (SMMR), a number of imagers have been launched along the years, including the Special Sensor Microwave Imagers (SSMI, SSMI/S), and the Advanced Microwave Scanning Radiometers (AMSR-E, AMSR2). The

latest instrument is the Global Precipitation Measurement (GPM) Microwave Imager (GMI) in 2014. Given the characteristics of the successive MW imagers available (see Table 1), similar frequencies are used across instruments, and have relatively close characteristics that could allow for a similar processing of the data starting from 1978.

We can divide the available instruments in 3 groups, based on the imaging frequencies used on each of them:

- Below 40 GHz: That particular set of channels is available onboard SMMR that flew from 1978 to 1987. On GMI, the

frequencies available are: 18.7 GHz (V,H), 23.8 GHz (V), and 36.5 GHz (V,H).

- Below 90 GHz: Available on the Special Sensor Microwave/Imager (SSM/I) (1987 to present) or on the Advanced Microwave Scanning Radiometer (AMSR-E/AMSR2) (2002 to present). Onboard GPM, the 89 GHz (V,H) is added.

- Below 190 GHz: This is for instance the case for the Special Sensor Microwave Imager Sounder (SSMIS) (2003 to present) or the GMI (2014 to present). In addition to all the above channels, 165.5 GHz (V,H) and 183.3 GHz (V)

channels are present.

| Imager | Channels (GHz) (Polarisation) | | | | | | Spatial Resolution | Viewing | Operating |
|---|---|---|---|---|---|---|---|---|---|
| | $\sim 18\,(V, H)$ | $\sim 23$ | $\sim 36\,(V, H)$ | $\sim 89\,(V, H)$ | 165.5 | 183.3 | (at 37 GHz) | angle | years |
| SMMR | 18.0 | 21.0 (V,H) | 37.0 | – | – | – | 29 km * 17 km | 50.2° | 1978–1987 |
| SSM/I | 19.4 | 22.2 (V) | 37.0 | 85.5 | – | – | 36 km * 24 km | 53.1° | 1987–2006 |
| AMSR-E | 18.7 | 23.8 (V,H) | 36.5 | 89.0 | – | – | 14 km * 9 km | 55.0° | 2002–2011 |
| SSMIS | 19.4 | 22.2 (V) | 37.0 | 91.6 | – | H,H | 44 km * 28 km | 53.1° | 2003– |
| AMSR2 | 18.7 | 23.8 (V,H) | 36.5 | 89.0 | – | – | 12 km * 7 km | 55.0° | 2012– |
| GMI | 18.7 | 23.8 (V) | 36.5 | 89.0 | V,H | V,V | 15 km * 9 km | 52.8° | 2014– |

**Table 1.** Characteristics of the MW imagers over the years.

All these instruments observe with a similar incidence angle at the surface (as a consequence the angular dependence is not to take into account as with sounders such as AMSU). The available frequencies are close (e.g., 37 GHz for SSM/I against 36.5 GHz for GMI and AMSR2) and with small differences in the operating bandwidth. Note that frequencies below 18 GHz are available for some of these instruments but they will not be considered as their sensitivity to clouds is very limited. In this

study, the passive microwave observations will come from GMI as it includes all the possible frequencies that we may want to use. Another benefit is that the GPM mission is not sun-synchronous and as a result, it covers the full diurnal cycle, whereas the other instruments are sun-synchronous with overpassing times at the equator in the morning and afternoon (SSMR, SSMI, SSMIS) or at mid-day and mid-night (AMSR-E and AMSR2). The cloud information comes from SEVIRI on board Meteosat:



it provides a cloud mask as well as a cloud classification. Rain is not detected per se, separately from the cloud: some clouds are likely to precipitate and the detection of these clouds will obviously include the detection of rain.

We first describe the data sets relevant for this study (Sect. 2). In Sect. 3, we will elaborate on the methodology. Results will be presented over land surfaces as well as over ocean (to illustrate the difference in behavior over these two surface types), insisting on the detection of the cloud contamination on the MW observations over land (Sect. 4). Sect. 5 concludes this study.

## 2   Data sources

The different data sources are described here, namely the SEVIRI cloud classification and the GMI brightness temperatures ($T_{bs}$). The steps to create a consistent data set are described, along with a preliminary analysis of the observations. Using ancillary data to help characterize the atmospheric and surface conditions related to the cloud occurrence (such as land surface emissivities atlases) could help the cloud detection but at the cost of increasing the complexity to apply it. For flexibility and convenience, the detection of the cloud contamination will be exclusively built from passive MW observations.

### 2.1   Cloud mask and classification from Meteosat SEVIRI

Meteosat is a geostationary satellite positioned over the equator. It covers mostly Africa, Southern America, Europe and the Middle East, from $\pm 60°$ latitude and $\pm 60°$ longitude. The SEVIRI channels on board Meteosat encompass the visible and infrared ranges (Schmid, 2000), with varying pixel sizes around $4\,\mathrm{km}^2$. Algorithms have been developed to provide cloud information, such as cloud top height, water content, and also cloud classification, every 15 minutes over the whole field-of-view (Derrien and Le Gléau, 2005).

The Climate Satellite Application Facilities (SAF) at the European Organisation for the Exploitation of Meteorological Satellites (EUMETSAT) provides daily data since 2004. We used the 2013 version of the SEVIRI cloud classification algorithm that provides a robust overview of the different cloud types that matter for VIS/IR observations. Using this classification, the goal is to improve our understanding of the MW interaction with clouds and to detect the cloudy situations that impact the MW. Six full days each month in 2015 provide 72 different daily situations that represent a large variation of the possible cloud types and surface conditions, covering the full diurnal and annual cycles. The cloud classes are described in Table 2. High semitransparent clouds are mostly cirrus of varying thickness, possibly over lower clouds. The fractional cloud class corresponds to cells that are only partly cloudy and to heterogeneous cloud cover. The other cloud types represent the continuum of possible cloud states, with varying opacity and height. Some of these clouds are likely to precipitate, and rain cases are naturally included in the database.

Figure 1 shows the latitudinal variation of the cloud types over land within the SEVIRI disk, for February and August 2015. The Inter-Tropical Convergence Zone (ITCZ) location changes between the two seasons, as expected. Over the mid-latitudes, the cloud frequency in February is higher than in August. The average relative frequency of each cloud type is displayed, showing that all cloud types are well represented.



| SEVIRI class description | Cloud type number |
|---|---|
| Cloud free land | 1 |
| Cloud free sea | 1 |
| Very low clouds | 2 |
| Low clouds | 3 |
| Medium clouds | 4 |
| High opaque clouds | 5 |
| Very high opaque clouds | 6 |
| High semitransparent thin clouds | 7 |
| High semitransparent meanly thick clouds | 8 |
| High semitransparent thick clouds | 9 |
| High semitransparent above lower clouds | 10 |
| Fractional clouds | 11 |

**Table 2.** Cloud classification from SEVIRI (Derrien and Le Gléau, 2005).

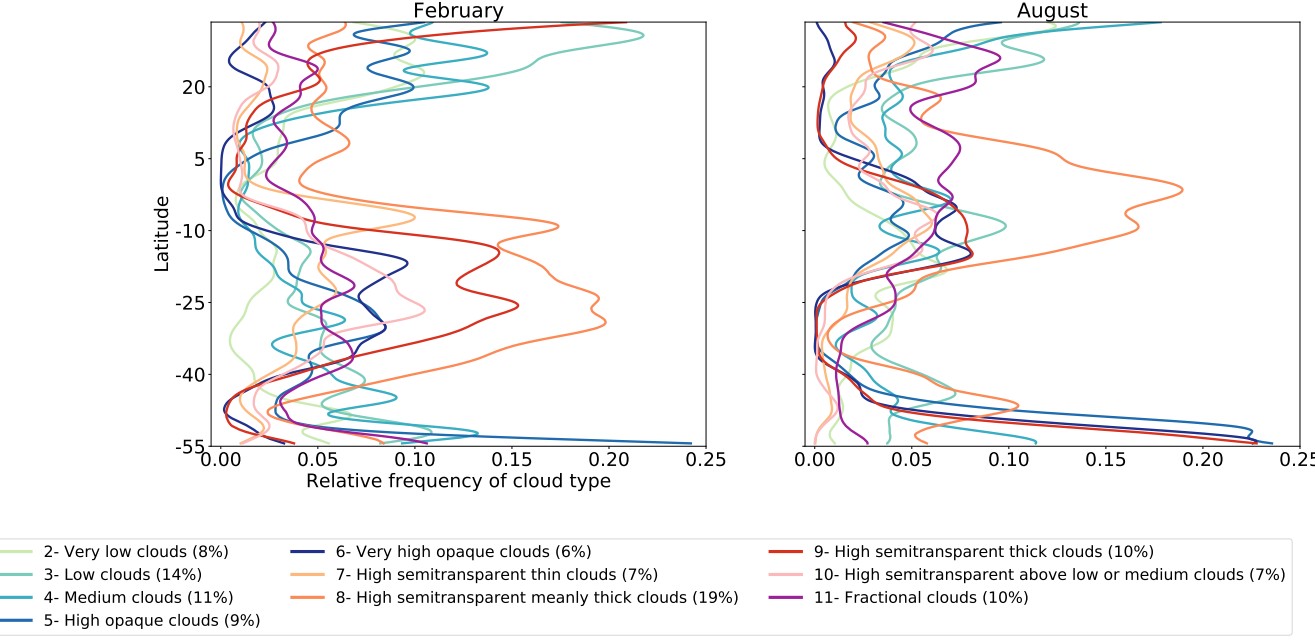

**Figure 1.** Relative frequency of cloud types as a function of latitude, for February (left) and August (right) 2015, over land within the SEVIRI disk. The frequency of each cloud type is indicated in the legend, averaged over these 2 months.





## 2.2 The passive microwave observations from GMI

GPM relies on several instruments to provide a precipitation evaluation around the globe. The GMI is on board the core GPM satellite. The satellite has a $65°$ inclination that allows a non sun-synchronous observation of the Earth. The available frequencies range from $10\,\mathrm{GHz}$ to $183\,\mathrm{GHz}$ (Hou et al., 2014). In this study, we use the corrected calibrated $T_{bs}$ created by the
level-1C algorithm.

GMI covers the full frequency range we want to analyse, with an incidence angle close to $53°$. In this study, different subsets of the channels will be tested, corresponding to the different channel ranges available on the instruments since 1978. In addition, it observes at different local times, limiting possible biases related to observations at specific times of the day. The GMI data from 2015 have been downloaded, for the 72 days corresponding to the SEVIRI selection.

## 2.3 Dataset preparation and preliminary analysis

The SEVIRI and GMI data have very different spatial and temporal resolutions. We need to find the closest matching observations and relocate them on a common grid for further processing. Each GMI observation has a time stamp that is used to find the closest SEVIRI scan. With SEVIRI data every $15\,\mathrm{min}$, there is a maximum of $7.5\,\mathrm{min}$ difference between GMI measurements and the corresponding SEVIRI classification. Grid cells with low quality flag are avoided, for both GMI and SEVIRI. Given
the spatial resolution, several SEVIRI cells will obviously fall in one GMI grid cell. Thus only GMI observations associated to a single SEVIRI class are kept, to reduce ambiguities in the training dataset. This does not mean that GMI cells with heterogeneous cloud cover will not be able to be classified: it just limits the confusion during the training phase. The grid cells located above $55°$ North and below $50°$ South are discarded: they are larger in size and are subject to more contamination by snow and ice. The GMI land mask is adopted to separate land and water bodies.

As a first analysis of the MW sensitivity to clouds, the distributions of the MW brightness temperatures ($T_{bs}$) are plotted in Fig. 2, for the different cloud types and for selected GMI frequencies, over ocean (left) and land (right). With increasing frequency, the atmospheric attenuation increases and the surface contribution to the signal decreases: the difference in the mean $T_{bs}$ between the ocean and land situations diminishes with higher frequencies. Differences in the signal received by the instrument, when it is not totally absorbed by the atmosphere, can be due to the cloud effect but can also be related to changes
in the surface properties (surface temperature of the ocean or land, wind speed at the ocean surface, soil moisture or vegetation density over land). Cloud types can be preferably associated to some environments, and the surface emissivity change with the surface conditions makes it difficult to find simple relationships between signals and cloud presence. In addition, water vapor modulates the MW signal, and this effect increases with frequencies in the window channels.

Over ocean up to $100\,\mathrm{GHz}$, the clouds are detectable and to some extent, their types can be distinguished: there is enough
contrast between the radiometrically cold ocean background and the cloud radiation. Above $100\,\mathrm{GHz}$, the surface contribution decreases drastically. The high opaque clouds can present low $T_{bs}$ (the long left tail of the histogram) that are related to the scattering by the cloud frozen phase.



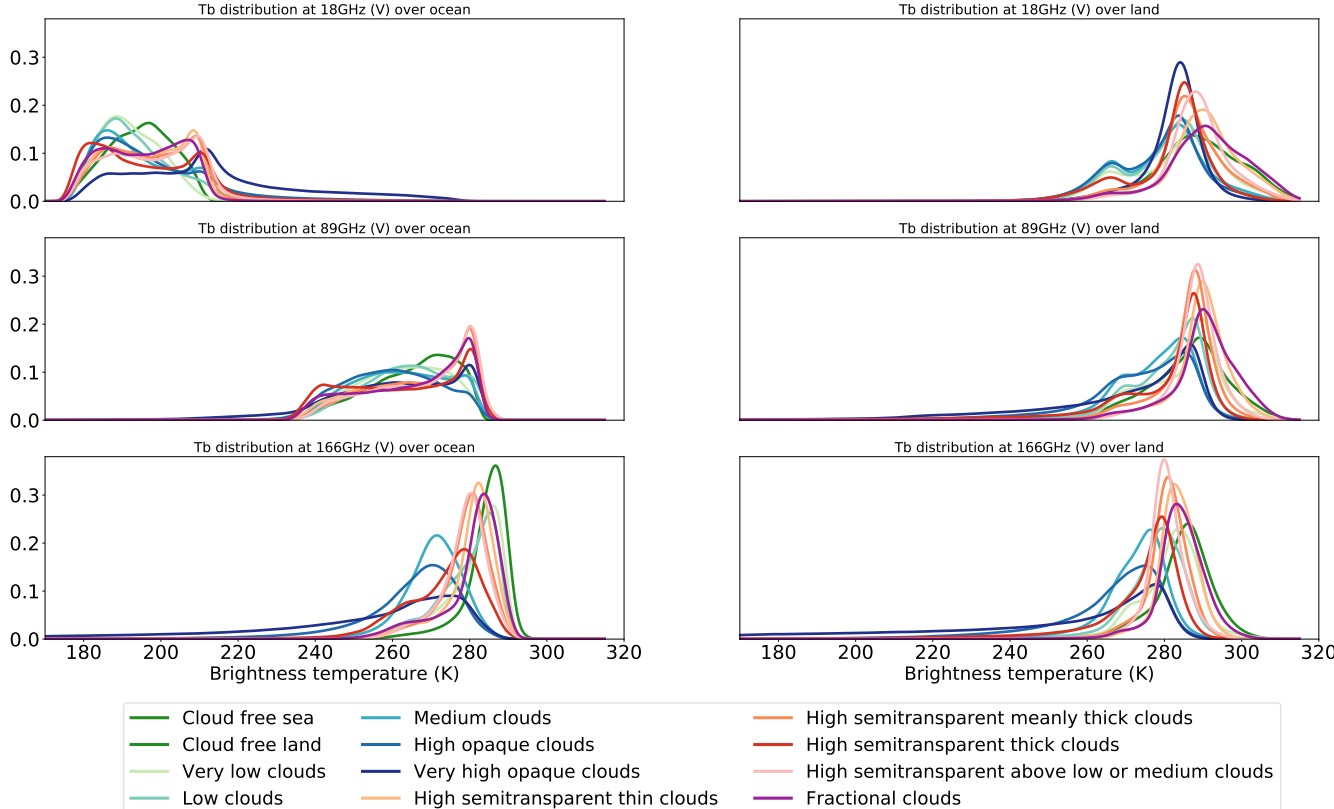

**Figure 2.** Probability distributions of the GMI observed $T_{bs}$ for various cloud types at $18\,\mathrm{GHz}$ (top), $89\,\mathrm{GHz}$ (middle), and $166\,\mathrm{GHz}$ (bottom) for the vertical polarization, over ocean (left) and land (right) from the filtered dataset.

Over land at $18\,\mathrm{GHz}$, the lowest peak in the histograms for most cloud types (around $265\,\mathrm{K}$) are likely related to the presence of water at the surface. Otherwise, at $18\,\mathrm{GHz}$, the histograms are very similar for all land situations, meaning that this frequency has a very limited sensitivity to the cloud presence and type. This can be seen as an asset for land surface characterization with these frequencies, as the signal will not be affected by the cloud presence. At high frequencies, the high opaque clouds present low $T_{bs}$ (the left tails of the histograms), due to the ice scattering in the clouds (as at $166\,\mathrm{GHz}$ over ocean). These opaque clouds will likely be detected over land with these high frequencies.

## 3 Method

Our goal is to detect cloud-contamination in the MW observations, over land. It is not at this stage to classify cloud types. It will nevertheless be interesting to analyse the effects of each cloud type in the different frequency domains. We focus here on the cloud detection for which a binary classification is required, but we will also experiment the cloud-type classification. Several methods are available, some are rule-based, mostly by using thresholds for the various cloud types (e.g., the SEVIRI



cloud algorithm by Derrien and Le Gléau (2005), or the cloud filter at $183\,\mathrm{GHz}$ from Buehler et al. (2007)). In this study, we use a statistical approach similar to the one presented in Aires et al. (2011).

## 3.1 The training and testing datasets

The training and testing datasets are constructed using the collocated GMI observations and SEVIRI cloud information. To cover the full diversity of cloud situation, a full year of data has been sampled with 72 days (Sect. 2). The SEVIRI acquisition disk excludes the high latitude regions and does not cover the full snow and ice free continents either. The development in this study will not be applicable to the snow and ice covered regions. However, it was shown in Aires et al. (2011) that the calibration of a cloud classification on the SEVIRI disk with MW observations can be extrapolated to the other continents and we are confident that the methodology will be applicable outside the SEVIRI disk, excluding the snow and ice regions. In the database, we impose that every cloud type is equally represented. This process ensures that the obtained classification will not be biased towards the most frequent cloud situations, disregarding the less frequent ones. We therefore sample the same number of clear and cloudy situations, with each cloud type equally represented in the cloudy part. This resulted in 1 million samples for each of the 10 cloud types, and 10 million cloud-free samples. For a cloud classification model, with 11 different possible output classes, the database is built with an similar repartition of classes, giving around 11M observations. The resulting databases are then randomly divided into the training (80%) and the testing (20%) datasets.

## 3.2 Statistical models

Several statistical models have been tested (e.g., tree-based or logistic regression, results not shown) but we kept a Neural Network (NN) classification, based on the MultiLayer Perceptron (MLP) (Rumelhart et al., 1986). MLP are universal nonlinear approximators, that can, given enough parameters, approximate any function (Hornik, 1991). The NN inputs are the MW channels, their number depending on the frequency ranges (5, 7, or 11). Five neurons (resp. 7 and 9) in the hidden layer are used. More neurons and larger network have been tested but they did not offer significant improvements in the resulting accuracy (results not shown). The output layer is composed of one binary output (for the cloud detection) or 11 binary outputs (for the cloud classification). The activation in the output layer is a $softmax$ function. The parameters of the MLP classifier are found during the learning stage where a cost function is minimized. Using a cost function measuring an error such as binary cross-entropy, the continuous output of the NN can represent the classification probability (Bridle, 1989). The prediction closest to 0 indicates a high probability of having a cloudy grid cell (respectively 1 for clear sky). The result of the continuous NN output can then be converted into a binary decision using a threshold to be defined. In the following graphs and results, if not otherwise specified, a decision threshold of 0.5 is applied to derive the binary classification. For multi-class outputs, the highest value among the output neurons is selected as the predicted class. The results are displayed showing the percentage of true positives (cloudy grid cells correctly detected), and true negatives (clear grid cells correctly predicted) from all the samples inside a test set.





## 4    Results

We first test the methodology over ocean, where clouds are expected to be easier to detect and quantify as we saw from the distributions in Fig. 2. It provides a testing ground for the method, before expanding it to the more difficult land case.

### 4.1    Detecting clouds over ocean

As described in Sect. 3.1, the database is created with an equal distribution of the cloud and clear conditions and a balanced repartition between the different cloud types. The cloud detection is evaluated for the three MW frequency ranges (all channels, below 100 GHz only, below 40 GHz only), and the results are presented in Table 3 for the test dataset. The cloud detection performs well over ocean, reaching at least 80% accuracy, even with a reduced number of channels. The low emissivity of the ocean ($\sim 0.5$) and its relative homogeneity makes it possible to correctly detect the cloud presence, even at low MW
frequencies.

| | All channels (%) | Below 100 GHz (%) | Below 40 GHz (%) |
|---|---|---|---|
| Clear grid cells correctly predicted | 91 | 89 | 89 |
| Cloudy grid cells correctly predicted | 81 | 74 | 72 |

**Table 3.** Results of a binary classification over the ocean for different MW frequency ranges.

These cloud detection results are very encouraging and the natural next step is to investigate a cloud classification over ocean, with the same MW frequency ranges. The dataset with all classes equally sampled is used, suitable for a multiclass classification. Similar NN schemes are implemented, with 11 possible output neurons representing the 10 cloud classes and the clear case, for the three frequency ranges. The confusion matrices (Fig. 3) display the results of the classification showing for
each class (y-axis), the percentage of the samples predicted to belong to one of the 11 possible classes (x-axis). The diagonal shows the correctly classified percentage for each cloud type. The highest accuracy is reached for the cloud-free ocean, for the three MW frequency ranges. It is occasionally confused with the high semitransparent meanly thick clouds (class 8) or the fractional clouds (class 11) as they may not significantly affect the measured $T_{bs}$. For opaque clouds (class 2-6), the highest percentages are near the diagonal: these cloud types are correctly classified or classified as a cloud with a similar altitude.
We see an increase in the detection of high opaque clouds (class 4/5) when the channel at 89 GHz is available. This can be explained by the increased detection of the ice content that this channels offers compared to lower frequencies. When all channels are available the discrimination between cloud layers is even easier resulting in a better classification. The high semi transparent clouds (class 7-8-9-10) are sometimes incorrectly classified as clear sky especially with only lower frequencies (due to channels less sensitive to high altitudes phenomena), or high semitransparent thick clouds (class 8) with higher frequencies which is expected given that they share similar properties (such as cloud height). Fractional clouds (class 11) are not well
classified, the predicted class being either cloud-free or high semitransparent (class 8).





## 4.2  Detecting clouds over land

A similar cloud detection method is applied over land. The NN classifiers are built, using the three different MW frequency ranges as inputs and with one output indicating the clear / cloudy probability.

The specifics of the model and database are dscribed in Sect. 3.1 and 3.2 Similar to Table 3 over ocean, Table 4 (top part) presents the accuracies reached over land by the three frequency ranges. The classification performance deteriorates compared to the ocean case, as expected. Nevertheless even for the worst case (with only 5 low frequency channels available) true positive and negative detections are close to 70%.

The result of the detection has been analyzed further, as a function of the cloud type (lower part of Table 4). Note that these are only a detail of the previous results (top part of Table 4) separated by each original cloud type. Large differences are observed between cloud types. For non semitransparent clouds, the higher the cloud the better the detection rate: this is directly related to the presence of ice in high clouds that can scatter the MWs. The higher the frequency, the better the detection of ice phase. Likewise, high semitransparent clouds can be detected only when they are thick enough.

| | All channels (%) | Below 100 GHz (%) | Below 40 GHz (%) |
|---|---|---|---|
| Clear grid cells correctly predicted | 83 | 73 | 69 |
| Cloudy grid cells correctly predicted | 77 | 73 | 73 |
| Very low clouds | 63 | 70 | 71 |
| Low clouds | 77 | 78 | 77 |
| Medium clouds | 92 | 85 | 83 |
| High opaque clouds | 97 | 85 | 83 |
| Very high opaque clouds | 98 | 92 | 90 |
| High semitransparent thin clouds | 59 | 56 | 54 |
| High semitransparent meanly thick clouds | 66 | 61 | 64 |
| High semitransparent thick clouds | 89 | 80 | 80 |
| High semitransparent above lower clouds | 84 | 74 | 71 |
| Fractional clouds | 53 | 48 | 46 |

**Table 4.** Top part: percentage of correct cloud detection from the test set over land. Lower part: Detail of the percentage of each cloud type predicted as cloudy. The results are presented for the three MW frequency ranges.





### 4.3 Detecting cloud contaminated microwave observations over land

The previous results showed that MWs cannot detect all clouds seen by VIS/IR measurements especially when only a subset of the frequencies is available. This behavior is actually very attractive for "all-weather" land surface applications with MWs. However, for accurate land surface characterization with MW, we need to identify the cloudy situations that are really

contaminating the MW. To that end we use the results from the previous model to select an appropriate definition of cloud-contamination in the MW. For all frequency ranges, high semitransparent thin clouds, high semitransparent meanly thick clouds, and the fractional clouds (i.e., classes 7-8-11), the classification accuracy is close to 50% similar to a random class assignment, meaning that these frequency ranges are not affected enough by these cloud types to be able to detect it. To focus on the clouds that do impact the MWs, we rebuild a training dataset, suppressing the 3 ambiguous classes previously men-

tioned (namely classes 7-8-11). The idea behind this new training database is that removing ambiguities at the learning stage will improve the classification. In other words, removing the ambiguous SEVIRI cloud types from the training database allows the model to ignore these phenomena mostly detected in VIS/IR. The lower sensitivity to clouds in MW is thus accounted for in the new training dataset. The results of this new classification are provided in Table 5, separately for the clear grid cells (SEVIRI class 1), for the cloudy grid cells with clouds that do contaminate the MW (the MW cloud-contaminated grid cells,

i.e., SEVIRI classes 2-3-4-5-6-9-10), and for the cloudy grid cells corresponding to the 3 cloud types difficult to detect from MW (the ambiguous grid cells, ignored in the training dataset, i.e., SEVIRI classes 7-8-11).

|  | All channels (%) | Below 100 GHz (%) | Below 40 GHz (%) |
|---|---|---|---|
| Clear cells correctly predicted | 88 | 77 | 71 |
| MW cloud-contaminated cells correctly predicted | 84 | 76 | 78 |
| Ambiguous grid cells predicted as MW cloud-contaminated | 49 | 43 | 52 |

**Table 5.** Classification results for the different clear and cloudy populations, for the three MW frequency ranges. See text for more details.

The results show that the clear sky detection increases and so does the detection of MW cloud-contaminated cells (84% with all frequencies) compared to the detection of cloudy cells in Table 4 (77% with all frequencies). This is expected as the ambiguous cases have been removed from the statistics; it is also consistent with the number of ambiguous cells (ignored in

the training datasets) that are predicted as MW cloud-contaminated by the new classification (close to 50% regardless of the frequency range).

The original output of the classification is not binary, but a number between 0 and 1 (see Section 3.2). In the results shown so far a decision threshold at 0.5 has been adopted to separate the two classes. Would it be possible to adjust this threshold




for a better detection of the cloud-contaminated observations? Figure 4 presents the outputs of the NN classifier, for the three populations previously defined in Table 5 and for each MW frequency range (Figure 4).

Figure 4 (top and middle panels) confirms that the clear grid cells and the MW cloud-contaminated grid cells are confidently classified, with very distinct output distributions for these two populations, 0 indicating a high confidence to be in the MW

cloud-contaminated class and 1 a high confidence to be in the clear class. Nevertheless, when channels above $100\,\mathrm{GHz}$ are not available, a non-negligible fraction of the clear grid cells population is classified between 0.1 and 0.4, meaning that the confidence in the prediction is lower. For the ambiguous cloud types that were ignored during the training (bottom panel), the distribution of the outputs covers a large range of values, traducing the uncertainty in the prediction. However, with the full frequency range there are a number of observations labelled as confidently contaminated (peak in low NN output values), this

can be expected due to the better sensitivity of the high frequency channels to thin clouds. Fig. 4 clearly shows that depending on the decision threshold selected for the NN output values, it is possible to filter out more or less ambiguous grid cells. It was so far set at 0.5, but it could be modified. The selection of this threshold should depend upon the frequency range and the application.

For instance, for land surface temperature estimates, the idea is to avoid the clouds that really affect the low microwave

Tbs (below $40\,\mathrm{GHz}$) that are used for the retrieval of this parameter (e.g., Prigent et al., 2016; Jiménez et al., 2017). Note however that this does not exclude the use of the higher frequencies for cloud-contamination detection, if these frequencies are also available. In addition, the interest of the MW for the land surface temperature estimation is to complement the infrared estimations that are not available under cloudy conditions: as a consequence, only the MW observations seriously cloud-contaminated should be detected, to maintain a quasi "all weather" coverage of the MW estimates while limiting erroneous

estimates under very cloudy / rainy situations. In that framework, the role of the cloud classification is to make sure the cloud-contaminated observations are correctly detected. The correct detection of the clear cases is of a lesser importance.

Figure 5 presents the percentage of MW observations predicted as cloud-contaminated, as a function of the threshold on the NN classifier output, for both the MW cloud-contaminated cases (the true positive, solid line) and the clear sky cases (the false positive, dash line). It shows that a threshold below 0.1 keeps the percentage of misclassified clear sky cases low (low

percentage of false positive). Combined with the results from Figure 4 (middle panel), a threshold at 0.05 and 0.01 could also be tested, to only classify the cloud-contaminated observations with a high degree of confidence.

A day of GMI observations, June 15th, 2015, is selected to illustrate the potential of the classification of the MW cloud contamination. Note that this day is not included in the training nor testing datasets previously used. For the three MW frequency ranges, the classification is applied with the selected thresholds (0.1, 0.05, 0.01). Table 6 provides the percentage of observa-

tions classified as cloud contaminated for each setup, along with the results from the Ferraro (1997) precipitation detection algorithms based on a decision tree and thresholds on channels. As expected, when the high frequency channels are included, the sensitivity of our methodology to the cloud contamination increases, as does the percentage of cloud-contaminated observations, with $\sim 10\%$ cloud contaminated observations for this frequency range. Note that for that day, the coincident SEVIRI observations are cloudy at 29%, i.e., three times more than the results from the high MW frequency range highest detection.

Using only frequencies below $40\,\mathrm{GHz}$, the percentage of cloud contaminated observations decreases. This illustrates the benefit



of using lower MW frequency channels for "all weather" land surface characterization, with a ratio of 4 between the number of contaminated observations when adding the 89 GHz to the frequencies below 40 GHz (using the 0.05 threshold). For all these thresholds/model combination the number of clear sky observations (according to SEVIRI) incorrectly flagged stays below 0.5% of all observations.

For comparison purposes, the Ferraro (1997) rain detection algorithms are also run, both the algorithm using the 85 GHz channel and the one limited to the frequencies below 40 GHz. The results in the second part of the table shows the number of observations that are flagged as precipitating. As expected the number of precipitating situations is lower than the number of contaminated MW observations. With models with channels above 40 GHz, more than 90% of the precipitating observations are detected by our method. The model with only channels below 40 GHz still retrieves more than 50% of the precipitating

observations when the 0.1 threshold is used.

    Nevertheless, depending on the applications and the degree of uncertainty required on the land surface product, if the full frequency range up to 100 GHz is available on the instrument, it can be relevant to use all the frequencies up to 100 GHz to filter out the cloud-contaminated grid cells, even if only the frequencies below 40 GHz are used in the retrieval of the parameter. As an example, if the land surface temperature is to be retrieved with very low uncertainty from SSM/I observations (an instrument

that has channels up to 90 GHz), it can be wise to use the full frequency range to detect the cloud contamination even if only the lower frequencies below 40 GHz are used in the retrieval.

| Threshold used | All channels (%) | Below 100 GHz (%) | Below 40 GHz (%) |
|---|---|---|---|
| 0.1 | 9.7 | 5.1 | 1.5 |
| 0.05 | 7.9 | 3.6 | 0.8 |
| 0.01 | 5.2 | 2.0 | 0.3 |
| Ferraro (1997) | – | 1.4 | 0.4 |

**Table 6.** Percentage of MW observations classified as cloud contaminated, for the three MW frequencies ranges, with different threshold on the NN classifier output. Results are presented for June 15th, 2015, over land surfaces within the SEVIRI disc. Second part of the table presents the percentage of observations detected as precipitating with Ferraro methods, with channels up to 100 GHz or only below 40 GHz.

    Now that we have an estimate of the number of points that are flagged by each model with different thresholds we can plot the global map of the location of these contaminated cells. Figure 6 shows the results for the 3 different frequencies group and with three thresholds applied. The threshold were chosen based on the results in table 6, to illustrate how different thresholds

might be applied to each model, while still providing coherent estimates of cloud-contaminated grid cells.

    In Fig. 6, models are applied to the data over land and sea, to create the 3 different maps. For each map a different threshold is applied, 0.1 with the lowest channels, 0.05 with channels up to 100 GHz and 0.01 with all channels available. The fourth subplot is the precipitating observations according to Ferraro's 89 GHz algorithm and the fifth shows the SEVIRI cloud type. We can analyse the output of this map:

– The agreement between models and the increased number of flagged points with more channels is clearly visible.



- In some areas, the cloudy grid cells do not appear to be detected, (i.e. red square area). When looking at the detail of the SEVIRI cloud types in that area we find out they are mostly fractional/semitransparent or low clouds, which explains the low contamination rate, according to our definition.

- In the pink area, we have a stronger detection of contaminated grid cells. Indeed the most represented cloud types are: High semi transparent thick clouds 23%, High semi transparent clouds above low or medium clouds 20%, and Very high opaque clouds 17%. All these cloud types are the ones that might affect the measurement the most.

- We find that the precipitating observations are correctly found under cloudy cells detected, but there are more cloud-contaminated observations.

This global application of our models shows the possible use with different frequencies range, to detect contaminated observations. Although adding more information by using the channels more sensitive to ice content leads to a better detection of cloud contamination, we show here that it is possible to filter out incorrect measurements even above land with a restricted number of channels. The threshold used here are coherent for the specific application shown here, with a low number of miss classified clear sky grid cells, and also with the real world occurrence of deep convective phenomena that contaminate the observations the most. Indeed the International Satellite Cloud Climatology Project (ISCCP) data shows that they have an average occurrence of 2.6% for deep convections that is of the same magnitude as our cloud index associated with the proposed thresholds (Rossow and Schiffer, 1999).

## 5 Conclusions

Passive microwave observations from satellites are less sensitive to clouds than visible / infrared measurements and can provide an almost "all weather" land surface characterization. However, cloud (and possible rain) can affect the microwave observations, even at frequencies below $40\,\mathrm{GHz}$. For accurate estimation of land surface parameters, cloud-contaminated MW observations have to be detected to avoid interpreting a cloud presence as a surface change.

A methodology has been developed to detect cloud contamination on passive MW observations, over land (except snow and ice covered areas). It is based on a NN classification, trained on collocated SEVIRI cloud types. The NN output indicates the probability of cloud-contamination in the MW signal, for a given MW frequency range. The cloud-contamination index is provided with values in the 0–1 range: the threshold applied to this index can be customized to fit the required application needed to flag out the contaminated observations. Although the target here is cloud detection over land surfaces, the model was also tested over the simpler case of detection over the ocean. The index confidence increased with the number of channels available, and performed better over the ocean as expected. In all cases, even with a reduced number of information over land, the model reaches an accuracy $\geq 70\%$ in detecting contaminated observations.

An example of a possible application of this cloud-contamination index to eliminate grid cells unsuitable for land surface temperature estimation was shown. The index proved useful to signal cloud contamination for this particular application and will soon be applied in the quality control of a long time record of land surface temperatures (Prigent et al., 2016). The land





surface temperature estimate is essentially based on passive microwave frequencies between 18 to 40 GHz, from a succession of satellite imagers since 1978 (SMMR, SSM/I, SSMIS). The first instrument only measured up to 36 GHz, contrarily to the last instruments. So far, the cloud and/or rain detection indices are based on indices related to channels around 85 GHz (Jiménez et al., 2017). This frequency is not available on board SMMR and the new methodology for the frequency range below 40 GHz

5    will be applied to the full data set, with possible comparisons with the method up to 100 GHz, when these channels are available. Overall these models can be applied globally and increase the detection of contaminating phenomena over the non-contaminating ones in the MW range.

*Data availability.*    The CLAAS-2 Cloud property dataset using SEVIRI - Edition 2 (CLAAS-2, DOI:10.5676/EUM_SAF_CM/CLAAS/V002) is publicly available from the Satellite Application Facility on Climate Monitoring (CM SAF). The GPM GMI_R Common Calibrated Bright-

10    ness Temperatures Collocated L1C 1.5 hours 13 km V05 (GPM_1CGPMGMI_R, DOI:10.5067/GPM/GMI/R/1C/05) is provided by NASA.

*Author contributions.*    All authors have been involved in interpreting the results, discussing the findings, and editing the paper. SF conducted the main analysis and wrote the draft of the paper. CJ, FA and CP provided guidance on using the data sets and expertise on analysing the results.

*Competing interests.*    The authors declare that they have no conflict of interest.

15    *Acknowledgements.*    This study was partly funded by the Centre National Centre National d'Études Spatiales (CNES, projects GPM-R and ISMAR) and the European Space Agency (ESA) project LST-CCI (Contract No. 4000123553/18/I-NB). Hervé Le Gléau et Gaelle Kerdraon (Centre de Météorologie Spatiale de Météo-France) are acknowledged by providing guidance to interpret the IR/VIS cloud classification, Martin Stengel (Deutscher Wetterdienst) by providing expertise on using the SEVIRI data, and Die Wang (Brookhaven National Laboratory) and Victorial Galligani (Centro de Investigaciones del Mar y la Atmósfera) by providing valuable comments to improve the manuscript.





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


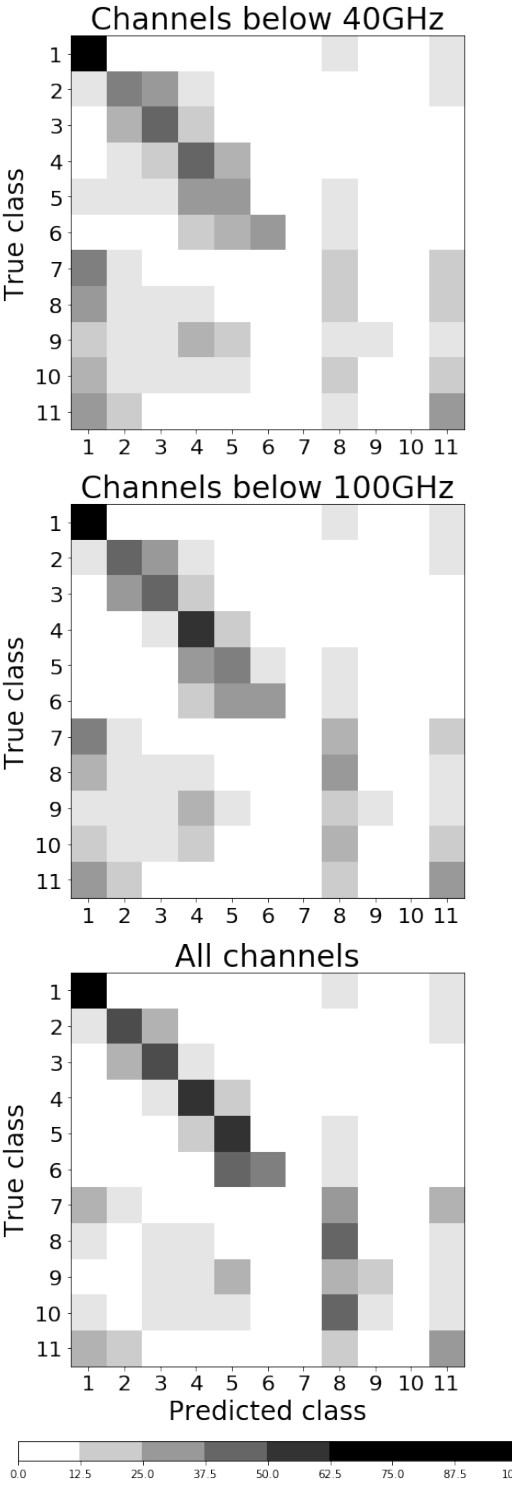

**Figure 3.** Confusion matrix over the ocean for the cloud type classification for the three MW frequency ranges: below 40 GHz (top), below 100 GHz (middle), and all channels (bottom). The cloud type numbers are detailed in Table 2.





**Figure 4.** Model output probability distributions for the clear grid cells (top), the MW cloud-contaminated grid cells (middle), and for the ambiguous grid cells (bottom), for the three MW frequency ranges. See text for more detail about the three populations.







**Figure 5.** Evolution of the percentage of MW observations correctly classified as cloud-contaminated (true positive, solid lines), and clear sky grid cells incorrectly classified as being contaminated (false positive, dashed line), as a function of the NN output threshold, for the three MW frequency ranges. Note that for this dataset half the observations are cloudy according to SEVIRI.





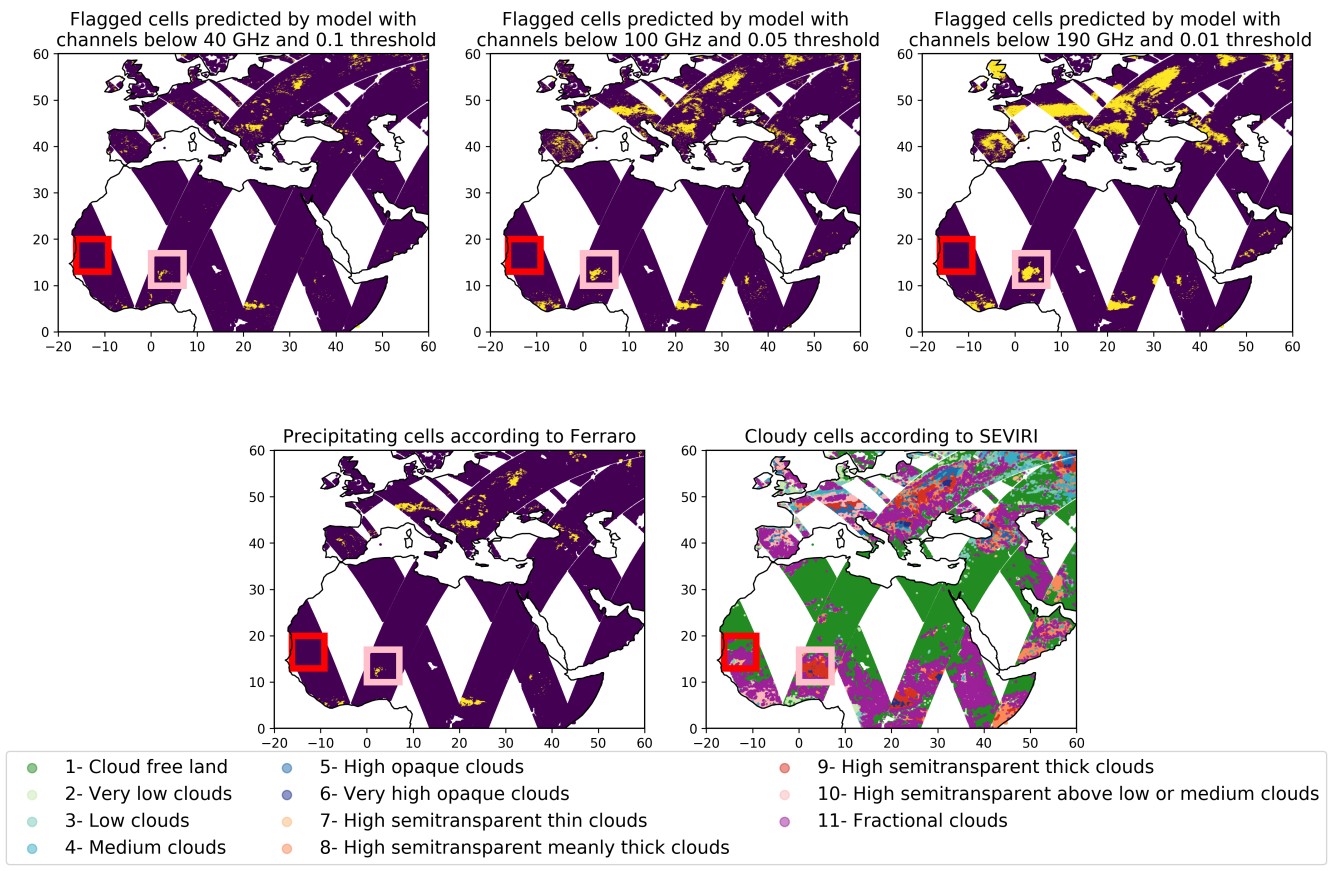

**Figure 6.** Maps showing the predicted grid cells flagged given by 3 models and thresholds with different channels available on June 15, 2015, compared to the detecting precipitating cells accoring to Ferraro (1997) and the cloudy classes from SEVIRI.