# Peer review of "Detecting cloud contamination in passive microwave satellite measurements over land"

_Atmospheric Measurement Techniques, 2018_

## Referee Comment (RC1) · Anonymous Referee #1 · 16 Nov 2018

General comments:

The manuscript addresses an interesting issue frequently ignored when retrieving Land Surface Temperature (LST) from satellite microwave (MW) data, namely a possibly non-negligible impact of cloud contamination on MW measurements. The developed neural network (NN) approach to cloud detection in MW data can use various channels available on MW imagers over the years, thereby making it applicable to historical and current satellite sensors. The training data consist of suitably filtered MW data (input) and SEVIRI cloud masks (desired output): ambiguous scenes were removed to avoid training the NNs with incorrect data, which improves the NNs ability to distinguish between cloud-free and cloud contaminated data. The reported results are in good agreement with previous findings, e.g. precipitation detected with Ferraro's (1977) method in MW data agrees well with the fraction of cloud contaminated MW observations detected with a NN and a suitably chosen threshold. The presented NN approach helps to ensure the accuracy of all-weather LST products retrieved from MW data.

The manuscript is well structured and generally well written, but needs proofreading. Particularly the abstract has to be improved (formulation, construction of sentences, spelling); however, the other parts of the manuscript also require another round of proofreading and stylistic improvements.

Specific comments:

- In the abstract it should say 'Meteosat Second Generation - ...'.

- Please check your use of the definite article.

- In modern English the word 'meanly' has not the meaning that is intended: consider using 'moderately' instead (many occurrences).

- I recommend to make the tables more reader-friendly, e.g. consider using different grey tones for the rows, swap columns 'Cloud type number' and 'SEVIRI class description' in table 2 (so the numbers are directly next to their respective labels), narrower tables 3, 4, 5, 6 (consider using background colours for the rows) .

- In table 4, please avoid two lines of text for the labels in the left column

- In table 5, consider providing the associated SEVIRI class numbers in the left column

Figure 1:

- Consider using different line styles for low, medium and high clouds

- Caption : 'The average frequency of each cloud type over these 2 months is indicated

[Figure]

in the legend.'

Figure 2:

- Insert tick marks for all x and y axes

- Consider using different line styles for low, medium and high clouds

Sections 3.1 and 3.2:

- You write about the data and the training, but there is no information on the NN software you used.

- I suggest providing figures showing the topologies of the 5-5-1 (and the 5-5-11?) network with appropriate labels - this would be more intuitive for the readers.

- Please write which NN training algorithm was used – simple Backprop or something more elaborate?

- Please provide information on the number of iterations / time required for the results to converge.

- Which stop criterion was used?

- Which hardware was used (e.g. standard office PC)?

Figure 5:

- Consider inserting a vertical line at x=0.1 to illustrate the threshold recommended in the text.

Figure 6:

- The subplots are too small. Consider showing two columns with the current top subplots as left column and the current bottom subplots as right column. Maybe also reduce geographical region. The colour scheme for SEVIRI clouds is too complex – at least I find it hard to see much in the LR subplot.

- Caption: please mention that the squares in the figures are explained in the text.

- The last sentence of the conclusions could be more specific (and up-beat).

---

## Referee Comment (RC2) · Anonymous Referee #2 · 4 Dec 2018

This manuscript describes a Neural Network(NN)-based method of detecting clouds from passive microwave observations. The NN is trained using co-located SEVERI cloud classifications and GMI observations. This method has several properties that make it useful for a variety of applications: the NN performs well over land, where physical methods of detecting clouds are more difficult than over water due to the more variable and heterogeneous surface, and the NN outputs a cloud probability metric that can be thresholded to screen more clouds (at the expense of increasing false detections) depending on the application.

I have a few minor comments that pertain to describing the method and its skill in more

detail. There are also some recommended technical corrections but none of these issues are major and the manuscript should be acceptable for publication after these revisions.

Minor Comments: 1. On page 3, there are some additional microwave imaging radiometers that could be included in your list. WindSat would fall into the group with channels < 40 GHz, and TMI could be listed under those with channels < 90 GHz.

2. Section 2.3 - I'm interested in the detail of how SEVIRI was mapped to GMI. Was the 36 GHz footprint of GMI used, or some other channel? The higher frequency (smaller) footprints might result in more homogeneous scenes for training but could also misrepresent the cloud type in the larger footprints.

3. Is there a reason that most of class 7-10 clouds are classified as class 8 by the NN? It seems as though these can't be distinguished from each other by the MW but are still being detected - you could group these classes together and training and confusion matrix would be more diagonal. I see these are excluded from the training altogether later, which is also logical since these optically thin ice clouds shouldn't have much impact on microwave brightness temperatures, and in fact any detection may be via correlation to upper level humidity inferred from the Tbs near 190 GHz.

4. Table 4: I think a measure of skill that evaluates the NN detection to random chance, such as the Heidke Skill Score, would be helpful here. For example, it is stated on Page 11 that the detection of classes 7-8-11 is similar to a random assignment, so there is no skill, while presumably, the detection of opaque clouds has higher skill. It would be helpful to see this for all classes to evaluate the relative detection capability.

Technical Corrections and Typographical Errors:

Page 2, Line 2: "spatial resolution" is erroneously repeated

Page 2, Line 8: "frequencies" should be "frequency"

Page 2, Line 23: "miss-interpreted" should be misinterpreted

Page 3, Line 2: "80's" should be "1980s"

Table 1: The 36 GHz channel on GMI is centered at 36.64 GHz (to avoid conflict with the Ka band radar)

Page 3, Line 16: "similar" should be "constant" in this context, I think

Page 4, Line 10: "emissivities" should be "emissivity"

Page 6, Line 4: for GMI, the level 1C product is internally calibrated (other sensors are intercalibrated to it), so I don't think "corrected" is needed here

Page 8, Line 5: "situation" should be "situations"

Page 14, Line 12: "miss classified" should be "misclassified"
* * *

---

## Author Comment (AC1) · 28 Dec 2018

The authors would like to express their gratitude for the reviewing done on the paper "Detecting cloud contamination in passive microwave satellite measurement over land". A detailed response to the reviewer comments on behalf of all authors can be found below.

**RESPONSE TO REVIEWER COMMENTS**

This manuscript describes a Neural Network(NN)-based method of detecting clouds from passive microwave observations. The NN is trained using co-located SEVERI

cloud classifications and GMI observations. This method has several properties that make it useful for a variety of applications: the NN performs well over land, where physical methods of detecting clouds are more difficult than over water due to the more variable and heterogeneous surface, and the NN outputs a cloud probability metric that can be thresholded to screen more clouds (at the expense of increasing false detections) depending on the application. I have a few minor comments that pertain to describing the method and its skill in more detail. There are also some recommended technical corrections but none of these issues are major and the manuscript should be acceptable for publication after these revisions.

Minor Comments: 1. On page 3, there are some additional microwave imaging radiometers that could be included in your list. WindSat would fall into the group with channels < 40 GHz, and TMI could be listed under those with channels < 90 GHz.

**R.** The mentioned MW imagers were not initially targeted for possible applications of our methodology, but given their frequency ranges and incidence angle our models should be applicable with minor differences. We are adding them as suggested.

2. Section 2.3 - I'm interested in the detail of how SEVIRI was mapped to GMI. Was the 36 GHz footprint of GMI used, or some other channel? The higher frequency (smaller) footprints might result in more homogeneous scenes for training but could also misrepresent the cloud type in the larger footprints.

**R.** The GMI data used is the 1C-R product, which provides GMI measured brightness temperatures with both low-frequency and high-frequency channels projected to a common scan centre position. This scan position is consistent with the resolution of the 89 GHz channel ( 4 km x 4km). The resolution of the 166 and 183 GHz channel is also close, while the coarser lower frequency channels are remapped to this resolution. As this common scan position is used to search for matches with the SEVIRI data, we can say that we are collocating at the resolution of the 89 GHz (and above) channels. Certainly, for the clouds not homogeneous at the scale of tens of kilometres the lower

the frequency the more likely the mismatch between the observed radiances and the SEVIRI cloud type. This adds some level of noise in the relationship between cloud types and radiances, and it is better explained now in the revised manuscript.

3. Is there a reason that most of class 7-10 clouds are classified as class 8 by the NN? It seems as though these can't be distinguished from each other by the MW but are still being detected - you could group these classes together and training and confusion matrix would be more diagonal. I see these are excluded from the training altogether later, which is also logical since these optically thin ice clouds shouldn't have much impact on microwave brightness temperatures, and in fact any detection may be via correlation to upper level humidity inferred from the Tbs near 190 GHz.

**R.** A possible explanation for the overrepresentation of class 8 as the predicted cloud type in figure 3, is that it is close to the average cloud cover for classes 7-10. The "meanly thick semitransparent cloud" corresponding Tb distribution possibly lies at the intersection of the "thin semitransparent" and "thick semitransparent" clouds ones. Nevertheless it is difficult to know which phenomenon leads to the distinction between cloud types. As seen in section 4.3 some cloud types are indeed difficult to detect, however they may share similar properties with other cloud types that led to this classification.

4. Table 4: I think a measure of skill that evaluates the NN detection to random chance, such as the Heidke Skill Score, would be helpful here. For example, it is stated on Page 11 that the detection of classes 7-8-11 is similar to a random assignment, so there is no skill, while presumably, the detection of opaque clouds has higher skill. It would be helpful to see this for all classes to evaluate the relative detection capability.

**R.** Following the reviewer suggestion we computed the Heidke Skill scores. The results are displayed in the table. The scores falling below 0.1 for class 7 and 11 illustrate well the lack of skill of our classifiers for these 2 classes. As we wanted a detection performing well for multiple frequency ranges and as the cloud types description covers

| Cloud Type | All channels | Channels (<100GHz) | Channels (<40GHz) | Mean Scores |
|---|---|---|---|---|
| 2 | 0.28 | 0.19 | 0.16 | 0.21 |
| 3 | 0.36 | 0.22 | 0.18 | 0.25 |
| 4 | 0.43 | 0.25 | 0.19 | 0.30 |
| 5 | 0.45 | 0.25 | 0.20 | 0.30 |
| 6 | 0.46 | 0.27 | 0.22 | 0.32 |
| 7 | 0.26 | 0.13 | 0.09 | 0.16 |
| 8 | 0.30 | 0.15 | 0.12 | 0.19 |
| 9 | 0.41 | 0.23 | 0.19 | 0.28 |
| 10 | 0.39 | 0.20 | 0.16 | 0.25 |
| 11 | 0.23 | 0.09 | 0.06 | 0.13 |

a continuum of possible cloud cover, the inclusion of class 8 along with the 2 previously mentioned made sense. These 3 classes indeed have the lowest skill score among all the cloud type.

Although we found the exercise of interest, we are not convinced that it shows anything particularly different from the analyses already included in the paper, so we prefer not to include them in the revised manuscript to not further complicate the discussion.

Technical Corrections and Typographical Errors: Page 2, Line 2: "spatial resolution" is erroneously repeated Page 2, Line 8: "frequencies" should be "frequency" Page 2, Line 23: "miss-interpreted" should be misinterpreted C2Page 3, Line 2: "80's" should be "1980s" Table 1: The 36 GHz channel on GMI is centered at 36.64 GHz (to avoid conflict with the Ka band radar) Page 3, Line 16: "similar" should be "constant" in this context, I think Page 4, Line 10: "emissivities" should be "emissivity" Page 6, Line 4: for GMI, the level 1C product is internally calibrated (other sensors are intercalibrated to it), so I don't think "corrected" is needed here Page 8, Line 5: "situation" should be "situations" Page 14, Line 12: "miss classified" should be "misclassified"

**R.** Technical suggestions and typographical errors corrected in the revised

---

## Author Comment (AC2) · 28 Dec 2018

The authors would like to express their gratitude for the reviewing done on the paper "Detecting cloud contamination in passive microwave satellite measurement over land". A detailed response to the reviewer comments on behalf of all authors can be found below.

**RESPONSE TO REVIEWER COMMENTS**

The manuscript addresses an interesting issue frequently ignored when retrieving Land Surface Temperature (LST) from satellite microwave (MW) data, namely a possibly

non-negligible impact of cloud contamination on MW measurements. The developed neural network (NN) approach to cloud detection in MW data can use various channels available on MW imagers over the years, thereby making it applicable to historical and current satellite sensors. The training data consist of suitably filtered MW data (input) and SEVIRI cloud masks (desired output): ambiguous scenes were removed to avoid training the NNs with incorrect data, which improves the NNs ability to distinguish between cloud-free and cloud contaminated data. The reported results are in good agreement with previous findings, e.g. precipitation detected with Ferraro's (1977) method in MW data agrees well with the fraction of cloud contaminated MW observations detected with a NN and a suitably chosen threshold. The presented NN approach helps to ensure the accuracy of all-weather LST products retrieved from MW data. The manuscript is well structured and generally well written, but needs proofreading. Particularly the abstract has to be improved (formulation, construction of sentences, spelling); however, the other parts of the manuscript also require another round of proofreading and stylistic improvements.

Specific comments: - In the abstract it should say 'Meteosat Second Generation - ...'.

**R.** Corrected.

- Please check your use of the definite article.

**R.** Done, thanks for pointing that out.

- In modern English the word 'meanly' has not the meaning that is intended: consider using 'moderately' instead (many occurrences).

**R.** The term "meanly" is only used in the label of one of the SEVIRI classes ("high semitransparent meanly thick clouds"). As it is the official designation of this type of cloud in the cloud classification we prefer to leave the label unchanged.

- I recommend to make the tables more reader-friendly, e.g. consider using different grey tones for the rows, swap columns 'Cloud type number' and 'SEVIRI class description' in table 2 (so the numbers are directly next to their respective labels), narrower tables 3, 4, 5, 6 (consider using background colours for the rows) .

**R.** The tables have been revised as suggested by the reviewer in the revised manuscript.

- In table 4, please avoid two lines of text for the labels in the left column

**R.** The table layout has been changed as suggested.

- In table 5, consider providing the associated SEVIRI class numbers in the left column

**R.** They have been added.

Figure 1: - Consider using different line styles for low, medium and high clouds

**R.** Figure modified as suggested by the reviewer.

- Caption : 'The average frequency of each cloud type over these 2 months is indicated in the legend.'

**R.** Corrected as indicated.

Figure 2: - Insert tick marks for all x and y axes

**R.** Added to the revised figure.

- Consider using different line styles for low, medium and high clouds

**R.** Figure modified as suggested by the reviewer.

Sections 3.1 and 3.2: - You write about the data and the training, but there is no information on the NN software you used.

**R.** We use the Keras library (https://keras.io, 2015).

- I suggest providing figures showing the topologies of the 5-5-1 (and the 5-5-11?) network with appropriate labels - this would be more intuitive for the readers.

**R.** We appreciate the suggestion, but given that we use a standard multilayer perceptron of one hidden layer well described in numerous papers and text books on neural networks, we deemed unnecessary to include them in the paper.

- Please write which NN training algorithm was used – simple Backprop or something more elaborate?

**R.** We use backpropagation (Rumelhart et al., 1986) to find the weights minimizing a binary cross-entropy loss function (Dreiseitl and Ohno-Machado, 2003).

- Please provide information on the number of iterations / time required for the results to converge.

**R.** The number of iterations is in the order of a few hundreds, with a largest number of iterations for the NNs having a largest number of input and outputs nodes, as expected.

- Which stop criterion was used?

**R.** Early-stopping, with the training haltered when the loss function is not decreasing during 5 consecutive epochs.

- Which hardware was used (e.g. standard office PC)?

**R.** A standard office laptop with 4 cores and 16 GB of RAM. This, plus all the previous details about the NN are now added in Section 3.2.

Figure 5: - Consider inserting a vertical line at x=0.1 to illustrate the threshold recommended in the text.

**R.** Figure in the revised manuscript changed as suggested.

Figure 6: - The subplots are too small. Consider showing two columns with the current top subplots as left column and the current bottom subplots as right column. Maybe also reduce geographical region. The colour scheme for SEVIRI clouds is too complex – at least I find it hard to see much in the LR subplot.

**R.** Figure updated as suggested.

- Caption: please mention that the squares in the figures are explained in the text.

**R.** Added.

- The last sentence of the conclusions could be more specific (and up-beat).

**R.** We updated the sentence as: "Overall the classification models developed in this study are potentially useful for numerous applications where it is of interest to identify possible cloud contaminations in observed MW radiances. For instance, in addition to the land surface temperature example, they can also be applied to select clear scenes for accurate MW emissivity estimation (Moncet et al., 2011), or to detect cloudy scenes for the analysis of deep convections (Prigent et al., 2011). "

**References**

Dreiseitl, S., and L. Ohno-Machado, Logistic regression and artificial neural network classification models: a methodology review, Journal of Biomedical Informatics, 35, 352-359, 2002.

Moncet, J.-L., Liang, P., Galantowicz, J. F., Lipton, A. E., Uymin, G., Prigent, C., and Grassotti, C., Land surface microwave emissivities derived from AMSR-E and MODIS measurements with advanced quality control, Journal of Geophysical Research: Atmospheres, 116, 2011.

Prigent, C., Rochetin, N., Aires, F., Defer, E., Grandpeix, J.-Y., Jimenez, C., and Papa, F., Impact of the inundation occurrence on the deep convection at continental scale from satellite observations and modeling experiments, Journal of Geophysical Research: Atmospheres, 116, 2011.

Rumelhart, D.E, G.E Hinton, and R.J. Williams, Learning representations by back-propagating errors, Nature, 323, 533-563, 1986.

---

## Author Response (AR1)

Paris,
29-01-2019

*«Detecting cloud contamination in passive microwave satellite measurements over land»*

Samuel Favrichon, Catherine Prigent, Carlos Jimenez, and Filipe Aires

Response to referees:

This document describes the modifications made to the article following the anonymous referrees comments. For each referee, detailed answers are provided for each comment. The modified manuscript with changed figures and the corrections highlighted is added in a second part of this document.

**Referee 1 response :**

General comments:

The manuscript addresses an interesting issue frequently ignored when retrieving Land Surface Temperature (LST) from satellite microwave (MW) data, namely a possibly non-negligible impact of cloud contamination on MW measurements. The developed neural network (NN) approach to cloud detection in MW data can use various channels available on MW imagers over the years, thereby making it applicable to historical and current satellite sensors. The training data consist of suitably filtered MW data (input) and SEVIRI cloud masks (desired output): ambiguous scenes were removed to avoid training the NNs with incorrect data, which improves the NNs ability to distinguish between cloud-free and cloud contaminated data. The reported results are in good agreement with previous findings, e.g. precipitation detected with Ferraro's (1977) method in MW data agrees well with the fraction of cloud contaminated MW observations detected with a NN and a suitably chosen threshold. The presented NN approach helps to ensure the accuracy of all-weather LST products retrieved from MW data. The manuscript is well structured and generally well written, but needs proofreading. Particularly the abstract has to be improved (formulation, construction of sentences, spelling); however, the other parts of the manuscript also require another round of proofreading and stylistic improvements.

*The authors would like to express their gratitude for the review done on the paper "Detecting cloud contamination in passive microwave satellite measurement over land". A detailed response to the reviewer comments can be found below, with page and line numbers referring to the change tracking manuscript attached to this document.*

Specific comments:
- In the abstract it should say 'Meteosat Second Generation - ...'.

*Corrected*

- Please check your use of the definite article.

*Done, thank you for pointing that out.*

- In modern English the word 'meanly' has not the meaning that is intended: consider using 'moderately' instead (many occurrences).

*The term "meanly" is only used in the label of one of the SEVIRI classes ("high semitransparent meanly thick clouds"). As it is the official designation of this type of cloud in the cloud classification we prefer to leave the label unchanged.*

- I recommend to make the tables more reader-friendly, e.g. consider using different grey tones for the rows, swap columns 'Cloud type number' and 'SEVIRI class description' in table 2 (so the numbers are directly next to their respective labels), narrower tables 3, 4, 5, 6 (consider using background colours for the rows) .

*The tables in the paper have been revised as suggested. The width of the tables will be updated in the final version of the paper to adapt to the journal required formatting.*

- In table 4, please avoid two lines of text for the labels in the left column

*The tables layout have been changed as suggested.*

- In table 5, consider providing the associated SEVIRI class numbers in the left column

*They have been added.*

Figure 1:
- Consider using different line styles for low, medium and high clouds

*Figure modified as suggested by the reviewer.*

- Caption : 'The average frequency of each cloud type over these 2 months is indicated in the legend.'

*Corrected as indicated.*

Figure 2:
- Insert tick marks for all x and y axes

*Done.*

- Consider using different line styles for low, medium and high clouds

*Figure modified as suggested by the reviewer.*

Sections 3.1 and 3.2:
- You write about the data and the training, but there is no information on the NN software you used.

*We use the Keras library (`https://keras.io, 2015`) running on top of a Tensorflow backend.*

- I suggest providing figures showing the topologies of the 5-5-1 (and the 5-5-11?) network with appropriate labels - this would be more intuitive for the readers.

*We appreciate the suggestion, but given that we use a standard multilayer perceptron of one hidden layer well described in numerous papers and text books on neural networks, we deemed unnecessary to include them in the paper.*

- Please write which NN training algorithm was used – simple Backprop or something more elaborate?

*We use backpropagation (Rumelhart et al., 1986) to find the weights minimizing a binary cross-entropy loss function (Dreiseitl and Ohno-Machado, 2003).*

- Please provide information on the number of iterations / time required for the results to converge.

*The number of iterations is of the order of a few hundreds, with a larger number of iterations required to converge for the NNs having a larger number of input and output nodes, as expected.*

- Which stop criterion was used?

*Early-stopping, with the training haltered when the loss function is not decreasing during 5 consecutive epochs.*

- Which hardware was used (e.g. standard office PC)?

*A standard office laptop with 4 cores and 16 GB of RAM. This, plus all the previous details about the NN are now added in Section 3.2 (p.7-8, l.29-5).*

Figure 5:
- Consider inserting a vertical line at x=0.1 to illustrate the threshold recommended in the text.

*Done.*

Figure 6:
- The subplots are too small. Consider showing two columns with the current top subplots as left column and the current bottom subplots as right column. Maybe also reduce geographical region. The colour scheme for SEVIRI clouds is too complex – at least I find it hard to see much in the LR subplot.

*Figure updated as suggested. The SEVIRI cloud colors were left as they were, but the reduced geographical region should improve readability.*

- Caption: please mention that the squares in the figures are explained in the text.

*Added.*

- The last sentence of the conclusions could be more specific (and up-beat).

*We updated the sentence (p.14, l.15) as: "Overall the models developed in this study can be applied globally in ice and snow free areas and are potentially useful for numerous applications where it is of interest to identify possible cloud contaminations in observed MW radiances. In addition to the land surface temperature example, this index can be useful to select clear scenes for accurate MW emissivity estimation (Moncet et al., 2011) or to detect cloudy scenes for the analysis of deep convections (Prigent et al., 2011)"*

Response to Anonymous referee 2:

This manuscript describes a Neural Network(NN)-based method of detecting clouds from passive microwave observations. The NN is trained using co-located SEVERI cloud classifications and GMI observations. This method has several properties that make it useful for a variety of applications: the NN performs well over land, where physical methods of detecting clouds are more difficult than over water due to the more variable and heterogeneous surface, and the NN outputs a cloud probability metric that can be thresholded to screen more clouds (at the expense of increasing false detections) depending on the application. I have a few minor comments that pertain to describing the method and its skill in more detail. There are also some recommended technical corrections but none of these issues are major and the manuscript should be acceptable for publication after these revisions.

*The authors would like to express their gratitude for the review done on the paper "Detecting cloud contamination in passive microwave satellite measurement over land". A detailed response to the reviewer comments can be found below, with page and line numbers referring to the change tracking manuscript attached to this document.*

Minor Comments: 1. On page 3, there are some additional microwave imaging radiometers that could be included in your list. WindSat would fall into the group with channels < 40 GHz, and TMI could be listed under those with channels < 90 GHz.

*The mentioned MW imagers were not initially targeted for possible applications of our methodology, but given their frequency ranges and incidence angle our models should be applicable with minor differences. We are adding them as suggested (p.3, l.5-23)*

2. Section 2.3 - I'm interested in the detail of how SEVIRI was mapped to GMI. Was the 36 GHz footprint of GMI used, or some other channel? The higher frequency (smaller) footprints might result in more homogeneous scenes for training but could also misrepresent the cloud type in the larger footprints.

*The GMI data used is the 1C-R product, which provides GMI measured brightness temperatures with both low-frequency and high-frequency channels projected to a common scan center position. This scan position is consistent with the resolution of the 89 GHz channel ( 4 km x 4km). The resolutions of the 166 and 183 GHz channels are also close, while the lower frequency channels coarser footprints are remapped to this resolution. As this common scan position is used to search for matches with the SEVIRI data, we can say that we are collocating at the resolution of the 89 GHz (and above) channels. Certainly, for the clouds not homogeneous at the scale of tens of kilometres the lower the frequency the more likely the possible intrusion of clouds creating a mismatch between the observed radiances and the SEVIRI cloud type. This adds some level of noise in the relationship between cloud types and radiances, and it is better explained now in the revised manuscript (section 2.2, p.5, l.14 and 2.3, p.6, l.5-10).*

3. Is there a reason that most of class 7-10 clouds are classified as class 8 by the NN?
It seems as though these can't be distinguished from each other by the MW but are still being detected - you could group these classes together and training and confusion matrix would be more diagonal. I see these are excluded from the training altogether later, which is also logical since these optically thin ice clouds shouldn't have much impact on microwave brightness temperatures, and in

fact any detection may be via correlation to upper level humidity inferred from the Tbs near 190 GHz.

*A possible explanation for the overrepresentation of class 8 as the predicted cloud type in figure 3, is that it is close to the average cloud cover for classes 7-10. The "meanly thick semitransparent cloud" corresponding Tb distribution possibly lies at the intersection of the "thin semitransparent" and "thick semitransparent" clouds ones. Nevertheless it is difficult to know which phenomenon leads to the distinction between cloud types. As seen in section 4.3 some cloud types are indeed difficult to detect, however they may share similar properties with other cloud types that led to this classification.*

4. Table 4: I think a measure of skill that evaluates the NN detection to random chance, such as the Heidke Skill Score, would be helpful here. For example, it is stated on Page 11 that the detection of classes 7-8-11 is similar to a random assignment, so there is no skill, while presumably, the detection of opaque clouds has higher skill. It would be helpful to see this for all classes to evaluate the relative detection capability.

*Following the reviewer suggestion we computed the Heidke Skill scores. The results are displayed below.*

| Cloud Type | All channels | Channels below 100GHz | Channels below 40GHz | Mean Scores |
|---|---|---|---|---|
| 2 | 0.28 | 0.19 | 0.16 | 0.21 |
| 3 | 0.36 | 0.22 | 0.18 | 0.25 |
| 4 | 0.43 | 0.25 | 0.19 | 0.30 |
| 5 | 0.45 | 0.25 | 0.20 | 0.30 |
| 6 | 0.46 | 0.27 | 0.22 | 0.32 |
| 7 | 0.26 | 0.13 | 0.09 | 0.16 |
| 8 | 0.30 | 0.15 | 0.12 | 0.19 |
| 9 | 0.41 | 0.23 | 0.19 | 0.28 |
| 10 | 0.39 | 0.20 | 0.16 | 0.25 |
| 11 | 0.23 | 0.09 | 0.06 | 0.13 |

*The scores falling below 0.1 for class 7 and 11 illustrate well the lack of skill of our classifiers for these 2 classes. As we wanted a detection performing well for multiple frequency ranges and as the cloud types description covers a continuum of possible cloud cover, the inclusion of class 8 along with the 2 previously mentioned made sense. These 3 classes indeed have the lowest skill score among all the cloud type.*

*Although we found the exercise of interest, we are not convinced that it brings much more information than the analyses already included in the paper. We prefer not to include it in the revised manuscript to avoid adding complexity to the discussion.*

Technical Corrections and Typographical Errors:
Page 2, Line 2: "spatial resolution" is erroneously repeated
Page 2, Line 8: "frequencies" should be "frequency"
Page 2, Line 23: "miss-interpreted" should be misinterpreted
Page 3, Line 2: "80's" should be "1980s"

Table 1: The 36 GHz channel on GMI is centered at 36.64 GHz (to avoid conflict with the Ka band radar)

Page 3, Line 16: "similar" should be "constant" in this context, I think

Page 4, Line 10: "emissivities" should be "emissivity"

Page 6, Line 4: for GMI, the level 1C product is internally calibrated (other sensors are intercalibrated to it), so I don't think "corrected" is needed here

Page 8, Line 5: "situation" should be "situations"

Page 14, Line 12: "miss classified" should be "misclassified"

*Technical suggestions and typographical errors are corrected in the revised manuscript.*

[revised manuscript text omitted]

---

## Author Response (AR2)

Paris,
06-02-2019

*«Detecting cloud contamination in passive microwave satellite measurements over land»*

Samuel Favrichon, Catherine Prigent, Carlos Jimenez, and Filipe Aires

Response to editor comments:

The following marked-up manuscript tracks the changes made in response to the editor's comments.

Specifically, the abstract was improved (p.1, l.0-5) :

*Remotely sensed brightness temperatures from passive observations in the microwave (MW) range are used to retrieve various geophysical parameters, e.g., near-surface temperature. Cloud contamination, although less of an issue at MW than at visible to infrared wavelengths, may adversely affect retrieval quality, particularly in the presence of strong cloud formation (convective towers) or precipitation. To limit errors associated with cloud contamination, we present an index derived from standalone MW brightness temperature observations which measures the probability of residual cloud contamination.*

Figure 2 was corrected to add axis labels, and figure 6 legend was clarified to improve readability.

The authors are grateful for the editor precise comments that were helpful to improve the manuscript.

[revised manuscript text omitted]